# Expectation–Evidence Prompting: Structuring Verification by Comparing Expected and Observed Evidence

## Abstract

Large language models (LLMs) often fail in factual verification due to hallucinations, unreliable truthfulness judgments, and opaque reasoning. We identify a structural limitation underlying these failures: LLMs directly compare claims with evidence without accounting for expected refutational alternatives. Specifically, we demonstrate that this omission leads to ambiguity in contradiction detection and unreliable abstention. Leveraging this observation, we introduce *Expectation–Evidence Prompting* (EEP), a cognitively inspired strategy that first generates supportive and refutational expectations from a claim and then aligns them with observed evidence. This bidirectional reasoning process enforces logical symmetry, reduces bias toward agreement, and provides a principled abstention mechanism. Across three fact-checking benchmarks—FEVER, PubHealth, and SciFact—EEP achieves consistent gains over strong prompting baselines, including an 86.3 macro-F1 on FEVER (+3.6 over Chain-of-Thought), 82.1 precision on PubHealth (highest among all methods), and 76.1 F1 on the Supports class in SciFact. These results demonstrate that embedding expectation–evidence alignment into prompt design yields more interpretable, robust, and trustworthy factual reasoning in LLMs.

## 1 Introduction

Large Language Models (LLMs) have made remarkable progress across diverse NLP tasks through large-scale pretraining and in-context learning. However, in fact-oriented scenarios they still exhibit persistent weaknesses. Outputs often contain hallucinations (fluent but factually incorrect generations Kim et al. (2024)), and they tend to show blind agreement with user claims Le et al. (2023). Their reasoning also becomes brittle when evidence is contradictory or incomplete Lee et al. (2023). These limitations are particularly critical in high-stakes domains such as healthcare or law, where unreliable factual reasoning can amplify misinformation Zhou et al. (2024).

Prompt engineering has emerged as a practical mitigation strategy through carefully designed instructions. Among popular methods, *Chain-of-Thought (CoT)* prompting Wei et al. (2022) improves interpretability by encouraging intermediate steps, but its reasoning paths remain free-form, variable, and dependent on few-shot exemplars—functioning as "few-shot black boxes." Other approaches, such as Self-Ask and Decomposition prompting, partially alleviate these issues but still rely on ad hoc reasoning traces without principled guarantees.

To overcome these structural weaknesses, we turn to insights from cognitive psychology, specifically the *Strategic Use of Evidence (SUE)* technique Hartwig et al. (2006); Granhag & Hartwig (2008). In human investigative settings, SUE improves deception detection by eliciting expectations about what evidence *should* exist if a statement were true or false, and then comparing those expectations against observed evidence to reveal inconsistencies. This reasoning pattern—*expectation construction → evidence comparison → conflict identification*—offers a transparent, symmetric, and cognitively plausible mechanism for verification. Inspired by this, we propose *Expectation–Evidence Prompting (EEP)*, a structured prompting framework for LLMs.

Rather than directly mapping claims to labels, EEP introduces an intermediate reasoning stage in which the model generates both *supportive expectations* (what evidence should be observed if the

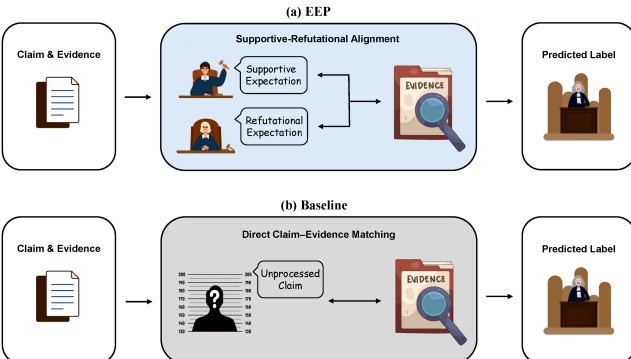

Figure 1: Comparison of (b) baseline and (a) EEP. Baseline methods directly match claims with evidence, overlooking refutational alternatives and causing ambiguity in contradiction detection. EEP generates supportive and refutational expectations and aligns them with evidence, enabling bidirectional reasoning, principled abstention, and more robust factual verification.

claim is true) and *refutational expectations* (what evidence should be observed if the claim is false) (see Figure 1). Observed evidence is then compared against these expectations, producing support and refutation scores that guide a structured three-way decision: support, refute, or abstain. This claim → expected evidence → observed evidence transformation yields three advantages over prior prompting methods: it enforces a fixed reasoning structure that is transparent and reproducible, it introduces logical symmetry by explicitly modeling both supportive and refutational pathways, reducing bias toward agreement, and it provides a principled abstention mechanism by design, enabling robust handling of ambiguous or insufficient evidence.

We evaluate EEP across three representative fact-verification benchmarks: FEVER Thorne et al. (2018), PubHealth Kotonya & Toni (2020), and SciFact Wadden et al. (2020). Experiments demonstrate that EEP consistently outperforms strong prompting baselines—Standard, CoT, Self-Ask, and Decompose prompting—particularly in macro-F1, contradiction detection, and abstention quality. Notably, EEP achieves an 86.3 macro-F1 score on FEVER (+3.6 over CoT), 82.1 precision on PubHealth (highest among all methods), and the strongest sensitivity to supportive evidence in SciFact. Despite these improvements, we also identify limitations: recall remains weaker in long-tail categories and cases requiring sophisticated refutational reasoning, pointing to the need for more fine-grained expectation modeling.

From a theoretical perspective, EEP can be understood as reducing the entropy of the reasoning space. By constraining LLMs to generate expectations under both the true and false hypotheses before evidence comparison, EEP contracts the hypothesis space, mitigating spurious correlations and enhancing the alignment between reasoning and decision outcomes. This structured approach thus bridges cognitive psychology principles with formal properties of interpretability in LLM prompting.

This work makes the following contributions:

- We introduce *Expectation–Evidence Prompting (EEP)*, a cognitively inspired framework that explicitly generates supportive and refutational expectations before evaluating evidence, aligning LLM reasoning more closely with human verification processes.

- We formalize EEP as a structured three-way decision process (support, refute, abstain), ensuring logical symmetry, introducing a principled abstention mechanism, and reducing entropy in the reasoning space.

- Through experiments on FEVER, PubHealth, and SciFact, we demonstrate that EEP consistently outperforms strong prompting baselines in macro-F1, contradiction detection, and abstention quality, establishing new robustness benchmarks in fact verification.

- We provide detailed error analysis showing that EEP excels in detecting supportive evidence but remains challenged by long-tail categories and refutational reasoning, identifying

promising directions for future extensions such as hybridizing with decomposition-based methods.

# 2 RELATED WORK

## 2.1 EVOLUTION OF PROMPT ENGINEERING: FROM DIRECT INSTRUCTION TO STRUCTURED REASONING

Prompt engineering has become a central mechanism for aligning human intent with LLM capabilities, with growing evidence that model performance is highly sensitive to prompt formulation, especially in fact verification tasks Liu et al. (2021).

The most basic form, *Standard Prompting*, relies solely on input–output pairs, mapping claims and evidence directly to labels without intermediate reasoning. While simple and widely adopted, it struggles with tasks requiring complex logical inference. To overcome these limitations, reasoning-enhanced strategies have been proposed. Among them, *Chain-of-Thought (CoT)* prompting encourages step-by-step reasoning before producing the final answer, improving performance on arithmetic, symbolic, and factual reasoning tasks Wei et al. (2022); Zhou et al. (2024). Building on this, Press et al. (2022) introduced *Self-Ask*, where the model generates and answers sub-questions, mimicking reflective cognition and proving effective in settings with incomplete or indirect evidence Hüyük et al. (2024). Another line of work, *Decompose Prompting*, breaks complex claims into smaller sub-tasks for separate verification before aggregation (Li et al., 2023), which has shown benefits in long or multi-step reasoning.

Despite these advances, existing strategies share key limitations: (1) they depend on models to autonomously generate intermediate steps, lacking fixed and transparent structure; (2) they often require task-specific re-design, limiting robustness; and (3) their reasoning processes remain engineering-driven demonstrations rather than cognitively grounded mechanisms. To address these gaps, we propose EEP, a strategy inspired by the SUE technique in cognitive psychology that models reasoning as "expectation construction–evidence comparison–conflict identification."

## 2.2 THE SUE TECHNIQUE IN INVESTIGATIVE PSYCHOLOGY

The Strategic Use of Evidence (SUE) technique is a validated interrogation method that improves deception detection by controlling the timing of evidence disclosure Hartwig et al. (2006); Granhag & Hartwig (2008). Its effectiveness lies in expectation violation: interviewees first provide accounts that implicitly reveal what evidence should exist if truthful, after which investigators disclose known evidence to identify inconsistencies (see Figure 2). Hartwig et al. reported a rise in detection accuracy from 56.1% to 85.4% using SUE compared to direct confrontation Hartwig et al. (2006).

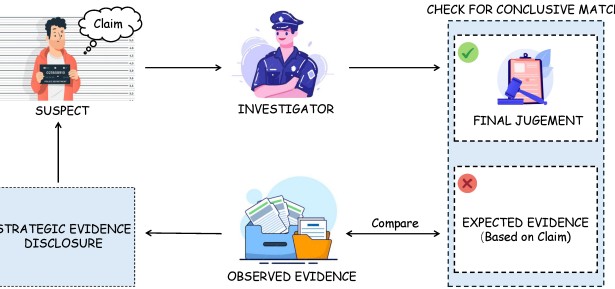

Figure 2: Cognitive structure of the SUE, where investigators compare suspect claims with expected and observed evidence.

This reasoning pattern—eliciting expectations, comparing them with reality, and detecting conflicts—maps naturally onto fact verification tasks. Unlike human subjects, claims in such tasks are static, but LLMs can be guided as "cognitive interviewers," first forming internal expectations and then aligning them with observed evidence. EEP adopts this cognitively inspired structure, moving beyond ad hoc reasoning traces toward a psychologically grounded prompting strategy.

### 2.3 Progress on Fact-Checking Benchmarks

Three benchmarks dominate fact verification research: FEVER, PubHealth, and SciFact. FEVER Thorne et al. (2018) established large-scale claim verification with multi-hop reasoning challenges, later enhanced through pretrained models Soleimani et al. (2020) and graph-based methods Zhong et al. (2020). With the rise of LLMs, prompt-based approaches and retrieval-augmented generation (RAG) frameworks have further advanced the field D'Monte et al. (2024); Salemi & Zamani (2024).

PubHealth Kotonya & Toni (2020) targets health claims with four-way classification (TRUE, FALSE, UNPROVEN, MIXTURE) and includes human-written explanations. Subsequent works explored explanation consistency and bilingual extensions Pavlopoulos et al. (2021); Vladika et al. (2024), though overall performance remains modest (F1 $\approx$ 67.5), underscoring the difficulty of uncertainty management in health fact-checking.

SciFact Wadden et al. (2020) focuses on scientific claims from biomedical literature, requiring reasoning across compressed abstracts. Studies have introduced evidence aggregation and graph reasoning Zhong et al. (2020); Wadden et al. (2021), while SciFact-Open highlighted retrieval challenges in open scholarly corpora. More recent efforts like SciFix Wadden et al. (2023) extend the task to claim correction. These benchmarks collectively reveal persistent issues: multi-sentence reasoning, cross-document integration, and lack of cognitively plausible inference. Our EEP framework addresses these gaps by embedding structured, expectation-based reasoning into LLM prompting.

## 3 Method

Existing prompting strategies for factual verification, such as Chain-of-Thought (CoT) or Self-Ask, rely on free-form reasoning chains that are entirely generated by the language model. While such strategies can increase interpretability by encouraging intermediate reasoning steps, they lack structural guarantees. The reasoning traces are unconstrained, variable across runs, and highly sensitive to prompt design. As a result, these methods often yield logically inconsistent arguments or shallow correlations between claims and evidence, rather than principled causal reasoning. This limitation mirrors the structural gap highlighted in the introduction: the absence of a fixed and transparent reasoning mechanism that systematically incorporates both supportive and refutational perspectives.

Expectation–Evidence Prompting (EEP) directly addresses this gap by introducing an intermediate stage of reasoning. Instead of comparing a claim $c$ with observed evidence $e$ in a single step, the model first generates *expected evidence* under two complementary hypotheses: the hypothesis that $c$ is true and the hypothesis that $c$ is false. These expectations create an explicit bidirectional space of reasoning, which is then matched against observed evidence (see Figure 3). This transformation converts fact verification into a structured consistency-checking problem, ensuring transparency and robustness. The design mirrors human cognitive strategies, such as those formalized in the Strategic Use of Evidence (SUE) technique, where hypotheses are tested against anticipated and actual outcomes to reveal conflicts.

### 3.1 Expectation–Evidence Framework

Formally, given a claim $c \in \mathcal{C}$ and a set of observed evidence passages $E_{\text{obs}} = \{e_1, e_2, \ldots, e_m\}$, EEP proceeds in three stages that together constitute a structured decision pipeline.

**1. Expectation Generation.** The model produces two sets of textual expectations, one under the assumption that the claim holds true and another under the assumption that it is false:

$$E_{\text{exp}}^+(c) = \{x_1^+, x_2^+, \ldots, x_k^+\}, \quad E_{\text{exp}}^-(c) = \{x_1^-, x_2^-, \ldots, x_\ell^-\}.$$

Here $E_{\text{exp}}^+(c)$ represents evidence that *should* be observed if $c$ were true, while $E_{\text{exp}}^-(c)$ represents evidence that *should* be observed if $c$ were false. The dual generation of supportive and refutational expectations enforces symmetry and ensures that contradiction detection is not reduced to absence of support.

**2. Evidence Comparison.** Next, the model evaluates how closely observed evidence aligns with either expectation set. This is operationalized using a semantic consistency function $\delta(\cdot, \cdot)$ that measures the degree of match between generated expectations and observed evidence. Two scores

Figure 3: Workflow of Expectation–Evidence Prompting (EEP). For each claim $c$, the model generates two types of expectations: $\mathcal{E}_{exp}(c)$ if the claim were true, and $\mathcal{E}_{exp}(\neg c)$ if the claim were false. The observed evidence $\mathcal{E}_{obs}$ is then compared against these expectations using a semantic alignment function $\delta(\cdot, \cdot)$. Based on threshold scores, the claim is classified as TRUE, FALSE, or OTHER.

are then defined:

$$S(c, E_{\text{obs}}) = \max_{x^+ \in E_{\text{exp}}^+(c)} \max_{e \in E_{\text{obs}}} \delta(x^+, e),$$

$$R(c, E_{\text{obs}}) = \max_{x^- \in E_{\text{exp}}^-(c)} \max_{e \in E_{\text{obs}}} \delta(x^-, e).$$

The support score $S$ measures the best alignment between supportive expectations and evidence, while the refutation score $R$ measures the best alignment between refutational expectations and evidence. By explicitly computing both, EEP prevents agreement bias and ensures that contradiction is tested through positive evidence of falsity rather than by elimination.

**3. Decision.** Finally, classification is performed by thresholding the support and refutation scores:

$$f(c, E_{\text{obs}}) = \begin{cases} \text{TRUE}, & \text{if } S(c, E_{\text{obs}}) \geq \tau_s \text{ and } R(c, E_{\text{obs}}) < \tau_r, \\ \text{FALSE}, & \text{if } R(c, E_{\text{obs}}) \geq \tau_r \text{ and } S(c, E_{\text{obs}}) < \tau_s, \\ \text{OTHER}, & \text{otherwise}, \end{cases}$$

where $\tau_s$ and $\tau_r$ denote the support and refutation thresholds, respectively. The OTHER class explicitly models the abstention option, capturing cases with insufficient or conflicting evidence. This abstention channel is crucial for real-world factual verification, where evidence is often incomplete or noisy.

## 3.2 SEMANTIC CONSISTENCY FUNCTION

The semantic consistency function $\delta$ can be instantiated in multiple ways, providing flexibility and allowing EEP to adapt across domains.

- **Implicit LLM reasoning.** Consistency is judged directly through language model generation. The prompt is designed to elicit a categorical decision (entail, refute, or unknown), effectively embedding $\delta$ within the generative process itself.

- **Embedding similarity.** Expectations and evidence passages are encoded into vector representations via a sentence encoder $\phi(\cdot)$, and similarity is measured using cosine distance:

$$\delta(x, e) = \frac{\langle \phi(x), \phi(e) \rangle}{\|\phi(x)\| \cdot \|\phi(e)\|}.$$

This instantiation is lightweight and interpretable, supporting fast large-scale inference.

- **Natural Language Inference (NLI).** A trained verifier estimates entailment probability:

$$\delta(x, e) = P(\text{entail} \mid x, e).$$

This instantiation benefits from supervised training on inference tasks and provides probabilistic grounding of consistency judgments.

These alternatives demonstrate that EEP is not tied to a single implementation but can flexibly incorporate existing semantic similarity or inference models.

### 3.3 WORKED EXAMPLE

Consider the claim $c$: *"The Eiffel Tower is taller than 500 meters."*

- Supportive expectation: *"Evidence should state that the Eiffel Tower's height exceeds 500 meters."*
- Refutational expectation: *"Evidence should mention that the Eiffel Tower is shorter than 500 meters."*
- Observed evidence: *"The Eiffel Tower stands at 324 meters, including its antenna."*

The alignment step yields a low support score and a high refutation score. Thresholding these values leads to the classification $f(c, E_{\text{obs}}) = \text{FALSE}$. This example highlights how explicit expectation generation enables the model to distinguish positive evidence of falsity from mere absence of support, thereby reducing ambiguity in contradiction detection.

### 3.4 ALGORITHMIC DESCRIPTION

---

**Algorithm 1:** Expectation–Evidence Prompting (EEP)

---

**Input:** Claim $c$, evidence passages $E_{\text{obs}}$
**Output:** Label $\in \{\text{TRUE}, \text{FALSE}, \text{OTHER}\}$
Generate expectations $E_{\text{exp}}^{+}(c)$ and $E_{\text{exp}}^{-}(c)$;
Compute support score $S(c, E_{\text{obs}}) = \max_{x^+, e} \delta(x^+, e)$;
Compute refutation score $R(c, E_{\text{obs}}) = \max_{x^-, e} \delta(x^-, e)$;
**if** $S \geq \tau_s$ **and** $R < \tau_r$ **then**
  $\llcorner$ **return** TRUE
**else if** $R \geq \tau_r$ **and** $S < \tau_s$ **then**
  $\llcorner$ **return** FALSE
**else**
  $\llcorner$ **return** OTHER

---

By explicitly modeling both supportive and refutational expectations, EEP enforces a form of *bidirectional reasoning*. This symmetry reduces bias toward agreement, strengthens contradiction detection, and ensures that abstention is principled rather than ad hoc. Unlike free-form prompting strategies, EEP produces a fixed and reproducible reasoning pathway that is both interpretable and empirically robust. Conceptually, EEP can be seen as contracting the hypothesis space of LLM reasoning: instead of evaluating claims in an unconstrained manner, the model is restricted to testing two opposing expectation sets against observed evidence. This contraction reduces reasoning entropy, suppresses spurious correlations, and enhances decision reliability. In doing so, EEP bridges cognitive psychology principles with formal properties of interpretability and robustness, offering a structured approach to factual verification that aligns with the central motivation of this work.

### 3.5 LEARNING WITH SUPERVISION

While EEP can be implemented as a zero-shot prompting strategy, it can also be extended into a trainable framework when labeled data are available. Specifically, we define a probabilistic model over labels $\mathcal{Y} = \{\text{TRUE}, \text{FALSE}, \text{OTHER}\}$.

**Logit Computation.** Given a claim $c$ and observed evidence $E_{\text{obs}}$, we compute support and refutation scores:

$$S(c, E_{\text{obs}}) = \max_{x^+, e} \delta(x^+, e), \qquad R(c, E_{\text{obs}}) = \max_{x^-, e} \delta(x^-, e).$$

We then define unnormalized logits:

$$z_{\text{True}} = S(c, E_{\text{obs}}), \quad z_{\text{False}} = R(c, E_{\text{obs}}), \quad z_{\text{Other}} = \gamma \cdot (1 - \max\{S, R\}),$$

where $\gamma$ is a scaling factor controlling abstention confidence.

**Softmax Prediction.** Final class probabilities are obtained via softmax:

$$p(y \mid c, E_{\text{obs}}) = \frac{\exp(z_y)}{\sum_{y' \in \mathcal{Y}} \exp(z_{y'})}, \quad y \in \mathcal{Y}.$$

**Loss Function.** Given a dataset $\mathcal{D} = \{(c_i, E_{\text{obs},i}, y_i)\}_{i=1}^{N}$, we train the model by minimizing the cross-entropy loss:

$$\mathcal{L}(\theta) = -\frac{1}{N} \sum_{i=1}^{N} \log p(y_i \mid c_i, E_{\text{obs},i}; \theta),$$

where $\theta$ denotes parameters of the scoring function $\delta(\cdot, \cdot)$ (e.g., an embedding model or NLI classifier).

**Regularization.** To encourage separation between support and refutation evidence, we introduce a margin-based regularization term:

$$\mathcal{R} = \frac{1}{N} \sum_{i=1}^{N} \max\left(0, \, m - |S(c_i, E_{\text{obs},i}) - R(c_i, E_{\text{obs},i})|\right),$$

where $m > 0$ is a margin hyperparameter. This penalizes cases where support and refutation scores are too close, thereby reducing indecision.

**Final Objective.** The overall training objective combines classification and regularization:

$$\mathcal{J}(\theta) = \mathcal{L}(\theta) + \lambda \mathcal{R},$$

where $\lambda$ controls the trade-off between accuracy and separation.

By training the $\delta$ function explicitly, EEP becomes more than a prompting heuristic: the learned support and refutation scores provide a transparent, interpretable measure of how well observed evidence aligns with hypothetical expectations. This preserves the cognitive motivation of the framework while making it compatible with standard supervised learning pipelines.

## 4 EXPERIMENTS

### 4.1 DATASETS

We evaluate EEP on three representative benchmarks. The FEVER dataset (Thorne et al., 2018), a large-scale benchmark for general-domain claim verification, is used in two settings: a balanced subset of 1,998 claims obtained through stratified random sampling (666 per class: SUPPORTS, REFUTES, NOT ENOUGH INFO) with a fixed random seed for reproducibility, and the full test set of 19,998 claims to assess scalability (detailed results in Appendix C). The PubHealth dataset (Kotonya & Toni, 2020), which contains health-related claims, exhibits strong label imbalance due to the scarcity of UNPROVEN examples; thus, we evaluate all methods on its complete test set of 1,214 claims to capture real-world distributional challenges. Finally, the SciFact dataset (Wadden et al., 2020) focuses on scientific claims grounded in biomedical literature. Its 452-claim test set is also imbalanced, with relatively few REFUTES and NOT ENOUGH INFO examples. To avoid sampling bias, we evaluate on the entire set, preserving the dataset's intrinsic challenges such as cross-sentence reasoning and the limited availability of negative evidence.

### 4.2 BASELINES

We compare EEP against four widely used prompting strategies for factual verification: *Standard Prompting*, *Chain-of-Thought (CoT)* (Wei et al., 2022), *Self-Ask* (Press et al., 2022), and *DE-COMP* (Khot et al., 2023). To ensure fairness, all methods use the same few-shot demonstrations (claim–evidence–label triples), deterministic decoding (temperature = 0), and consistent prompt templates (see Appendix C, Table 4 and 5).

**Standard.** Few-shot prompting with direct claim-to-label mapping; the model outputs only the final classification without intermediate reasoning.

**CoT.** Prompts include few-shot exemplars with step-by-step reasoning. The model generates free-form intermediate steps before predicting the label. Reasoning quality is unconstrained.

**Self-Ask.** The model generates clarifying sub-questions and corresponding answers in a dialogue style before outputting a label. Effectiveness depends on the quality of generated questions.

**DECOMP.** Claims are decomposed into multiple sub-questions (Q) with corresponding answers (A) derived from evidence, followed by a final decision. Performance is sensitive to both decomposition quality and answer accuracy.

### 4.3 MODEL AND IMPLEMENTATION DETAILS

All experiments are conducted using *GPT-3.5 Turbo* via the official ChatCompletion API. For reproducibility, decoding temperature is fixed to 0 (greedy decoding). Each method is implemented as a multi-turn, Socratic-style prompt template that captures its respective reasoning mechanism: direct input–output mapping for Standard, free-form reasoning for CoT, clarifying Q&A for Self-Ask, decomposition for DECOMP, and expectation–evidence comparison for EEP. Only the final classification labels are retained for evaluation, ensuring consistency across methods by discarding intermediate reasoning traces.

Table 1: Results on the FEVER dataset (labels: SUPPORTS / REFUTES / NEI). The best result for each column is highlighted in bold.

| Method | F1-Score (per class) | | | Macro-Averaged Metrics | | | N |
|---|---|---|---|---|---|---|---|
| | SUPPORTS | REFUTES | NEI | Precision | Recall | F1 | |
| EEP | 90.2 | 80.3 | 88.3 | 88.2 | 86.7 | 86.3 | 1998 |
| Standard | 84.9 | 74.8 | **95.0** | 87.6 | 85.6 | 84.9 | 1998 |
| CoT | 87.1 | 71.5 | 89.4 | 85.7 | 83.8 | 82.7 | 1998 |
| Self-Ask | 85.4 | 73.4 | 80.7 | 81.6 | 80.2 | 79.8 | 1998 |
| DECOMP | 90.3 | 68.6 | 81.5 | 85.4 | 81.1 | 80.2 | 1998 |
| EEP-Full | **90.5** | **81.3** | 88.9 | **88.7** | **87.3** | **86.9** | 19998 |

Table 2: Results on the PubHealth dataset (labels: TRUE, FALSE, UNP = Unproven, MIX = Mixture). The best result for each column is highlighted in bold.

| Method | F1-Score (per class) | | | | Weighted-Averaged Metrics | | | N |
|---|---|---|---|---|---|---|---|---|
| | TRUE | FALSE | UNP | MIX | Precision | Recall | F1 | |
| EEP | **84.6** | 66.0 | **19.4** | 39.1 | **82.1** | 64.7 | 70.4 | 1214 |
| Standard | 77.5 | 71.0 | 18.7 | **42.3** | 80.1 | 64.3 | 68.7 | 1214 |
| CoT | 77.4 | 70.5 | 12.6 | 40.4 | 78.4 | 63.0 | 68.0 | 1214 |
| Self-Ask | 65.3 | 70.0 | 11.2 | 30.3 | 76.1 | 53.5 | 60.2 | 1214 |
| DECOMP | 81 | **73.4** | 16.9 | 39.6 | 80.4 | **65.4** | **70.9** | 1214 |

Table 3: Results on the SciFact dataset (labels: SUPPORTS / REFUTES / NOINFO). The best result for each column is highlighted in bold.

| Model | F1-Score (per class) | | | Weighted-Averaged Metrics | | | N |
|---|---|---|---|---|---|---|---|
| | SUPPORTS | REFUTES | NOINFO | Precision | Recall | F1 | |
| EEP | **76.1** | 63.8 | 63.0 | 72.9 | 69.0 | 69.5 | 452 |
| Standard | 69.1 | 44.1 | 14.9 | 59.0 | 56.9 | 51.0 | 452 |
| CoT | 73.2 | 46.5 | 60.7 | 69.7 | 65.5 | 62.9 | 452 |
| Self-Ask | 66.5 | 59.2 | 55.7 | 68.2 | 61.1 | 61.8 | 452 |
| DECOMP | 76.0 | **67.0** | **66.9** | **73.4** | **71.0** | **71.3** | 452 |

## 4.4 RESULTS AND DISCUSSION

On the FEVER dataset, EEP achieved the highest macro-F1 score (86.3), with precision and recall of 88.2 and 86.7, respectively. This represents a clear improvement over Standard (84.9), CoT (82.7), Self-Ask (79.8), and DECOMP (80.2). At the category level, EEP obtained F1 scores of 90.2 (SUPPORTS), 80.3 (REFUTES), and 88.3 (NEI), reflecting balanced classification across labels. When scaled to the full FEVER test set of 19,998 claims, EEP's macro-F1 increased to 86.9, with precision and recall remaining stable, demonstrating robustness and scalability in large-scale inference (Table 1).

On PubHealth, EEP achieved a weighted F1 of 70.4, outperforming Standard (68.7), CoT (68.0), and Self-Ask (60.2), while remaining slightly below DECOMP (70.9). Notably, EEP attained the highest precision among all methods (82.1), underscoring its reliability in medical claim verification. Class-level analysis shows strong performance on TRUE (84.6) and UNPROVEN (19.4), competitive results on MIXTURE (39.1), but weaker performance on FALSE (66.0). This shortcoming likely reflects both the limited size of the FALSE class and the semantic difficulty of refutational reasoning, which requires integrating diverse counter-evidence (Table 2).

On SciFact, EEP obtained a weighted F1 of 69.5, outperforming Standard (51.0), CoT (62.9), and Self-Ask (61.8), though slightly trailing DECOMP (71.3). Within categories, EEP achieved the highest F1 in SUPPORTS (76.1), demonstrating strong sensitivity to positive evidence. However, it lagged behind DECOMP in REFUTES (63.8 vs. 67.0) and NOINFO (63.0 vs. 66.9), highlighting persistent challenges in handling refutational reasoning and missing evidence (Table 3).

Taken together, these findings demonstrate that EEP offers consistent gains over Standard, CoT, and Self-Ask, achieves state-of-the-art performance on FEVER, and remains competitive with DE-COMP on PubHealth and SciFact. The improvements can be attributed to EEP's explicit modeling of supportive and refutational expectations, which reduces ambiguity in evidence comparison and strengthens contradiction detection. At the same time, its limitations are evident: weaker performance in long-tail categories and cases requiring strong counter-evidence reasoning. These observations point to promising future directions, including hybridizing EEP with decomposition frameworks, generating more fine-grained counterfactual expectations, and developing strategies to better handle evidence-scarce scenarios. Overall, EEP demonstrates that embedding structured, cognitively inspired reasoning into prompt design enhances both interpretability and robustness in factual verification.

## 5 CONCLUSION

We introduced *Expectation–Evidence Prompting (EEP)*, a cognitively inspired prompting framework that imposes structure on factual verification. Instead of relying on unconstrained reasoning, EEP simulates the human process of constructing both supportive and refutational expectations and aligning them with observed evidence. This expectation–evidence comparison enforces bidirectional reasoning, reduces ambiguity in contradiction detection, and provides a principled abstention mechanism. Across three representative benchmarks—FEVER, PubHealth, and SciFact—EEP consistently outperformed widely used prompting strategies such as Standard, Chain-of-Thought, and Self-Ask, while remaining competitive with decomposition-based approaches. The method achieved state-of-the-art performance on FEVER, demonstrated the highest precision on PubHealth, and showed strong sensitivity to supportive evidence in SciFact. These results highlight EEP's strengths in precision, robustness, and interpretability, confirming its value as a structured alternative to free-form reasoning. Our analysis also revealed important limitations. EEP underperforms in long-tail categories and scenarios requiring complex refutational reasoning or reasoning under scarce evidence. These weaknesses underscore promising avenues for future work, such as hybridizing EEP with decomposition frameworks, generating more fine-grained counterfactual expectations, and extending the framework to tasks that demand stronger counter-evidence modeling.

## ETHICS STATEMENT

This work aims to improve factual verification in LLMs, with the goal of reducing hallucinations in domains such as healthcare and science. All experiments use publicly available datasets (FEVER,

PubHealth, SciFact), and no personal or sensitive data are involved. We caution that fact-checking systems may still be misused, and recommend human oversight in high-stakes applications.

## REPRODUCIBILITY STATEMENT

We have made all code, prompt templates, and evaluation results publicly available at the anonymous repository: `https://anonymous.4open.science/r/EEP-C7C7`. Our experiments were conducted on GPT-3.5 Turbo with temperature = 0 to ensure deterministic outputs. Dataset splits, prompts, and evaluation metrics are fully documented in the appendix. Together, these resources enable full replication of our reported results.

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

APPENDIX

## APPENDIX A. CODE AND DATA AVAILABILITY

All code, prompt templates, and evaluation results are available at: `https://anonymous.4open.science/r/EEP-C7C7`

## APPENDIX B. USE OF LLMs

This work makes direct use of large language models (LLMs) as part of the experimental evaluation. Specifically:

Models used. All experiments were conducted using OpenAI's GPT-3.5 Turbo model, accessed via the official API. No other proprietary or fine-tuned LLMs were used.

Purpose of use. The LLM served as the target model to be prompted for fact verification across the FEVER, PubHealth, and SciFact datasets. Our proposed Expectation–Evidence Prompting (EEP) framework was implemented entirely through prompting without additional training.

Configuration. We used deterministic decoding (temperature = 0) to ensure reproducibility. Few-shot examples and structured templates were provided as input prompts, as documented in Appendix B and Appendix C.

Impact and limitations. Results may be influenced by the inherent biases or knowledge cutoff of GPT-3.5 Turbo. To mitigate these effects, we standardized prompts, fixed seeds for dataset sampling, and released all code and templates for transparency.

No ghostwriting. The LLM was not used for writing or editing the manuscript; all text was authored by the research team.

## APPENDIX C. COMPACT PROMPT TEMPLATES

Table 4: Compact prompt templates, showing only [TD] Task Declaration, [FS] Few-shot Examples, and [SR] Structured/Unstructured Reasoning. Full templates with [DI] Data Input and [FL] Final Label are provided in Tables 6, 7, 8, 9, and 10.

| Method | Compact Template (TD / FS / SR) | Explanation |
|---|---|---|
| **EEP** | **[TD]** You are a fact-checking agent. Use the Expectation–Evidence Prompting (EEP) strategy to analyze claims. Your task is... 
 **[FS]** Here are examples: [examples] 
 **[SR]** Structured 4-step reasoning (expect true evidence / expect false evidence / compare with actual / decide label). | **Structured** Expectation vs. contradiction |
| **Standard** | **[TD]** You are a fact-checking agent. Use the Standard (answer-only) strategy to analyze claims. Your task is... 
 **[FS]** Here are examples: [examples] 
 **[SR]** No reasoning; directly output the final label. | **Unstructured** Direct answer only |
| **CoT** | **[TD]** You are a fact-checking agent. Use the Chain-of-Thought (CoT) strategy to analyze claims. Your task is... 
 **[FS]** Here are examples: [examples] 
 **[SR]** Step-by-step reasoning on the evidence. | **Unstructured** CoT |
| **Self-Ask** | **[TD]** You are a fact-checking agent. Use the Self-Ask prompting strategy to analyze claims. Your task is... 
 **[FS]** Here are examples: [examples] 
 **[SR]** Iterative follow-up Q&A until decision. | **Unstructured** Iterative Q&A |

**Table 4 (continued)**

| Method | Compact Template (TD / FS / SR) | Explanation |
|--------|----------------------------------|-------------|
| **DECOMP** | **[TD]** You are a fact-checking agent. Use the Decompose-Then-Reason (DE-COMP) strategy to analyze claims. Your task is... 
 **[FS]** Here are examples: [examples] 
 **[SR]** Break into sub-questions, answer sequentially, then synthesize. | **Unstructured** Decomposition → synthesis |

Table 5: Illustrative reasoning traces for a single claim across prompting methods. Only EEP represents structured prompting (explicit step-by-step instructions). The other methods (CoT, Self-Ask, DECOMP, Standard) are unstructured prompting, where the reasoning process is flexible to the model.

**Claim:** "Drug effective in smoking cessation studies."
**Evidence:** Two well-designed randomized, double-blind trials (N=2000) evaluated abstinence rates. Continuous ... methods.

| Method | Example Reasoning (excerpt) |
|--------|------------------------------|
| **EEP** | Step 1: If the claim is true, there should be evidence that the drug achieved significantly higher quit rates than placebo or control groups in randomized controlled trials, and that its effectiveness was recognized. 
 Step 2: If the claim is false, there should be evidence that the drug showed no significant difference compared to placebo or control groups, that its effect was uncertain, or that it had notable adverse effects. 
 Step 3: The actual evidence shows the drug outperformed placebo in trials (matching Step 1), but lacked comparisons with non-drug methods (partly matching Step 2), so the conclusion is partial support. 
 Step 4: Based on the above, the final label is: MIXTURE |
| **Standard** | Final label: MIXTURE |
| **CoT** | Reasoning: The evidence supports drug effectiveness relative to other drugs and placebo but lacks comparison to non-drug methods, so support is partial. 
 Final label: MIXTURE |
| **Self-Ask** | Are follow-up questions needed: Yes. 
 Follow up: How does the evidence indicate the drug's effectiveness? 
 Intermediate answer: The evidence shows higher continuous abstinence rates in the drug groups compared with placebo, indicating effectiveness. 
 Follow up: Does the evidence also provide comparisons with non-drug interventions? 
 Intermediate answer: The evidence states that no quantitative comparison with non-drug methods was included. 
 So the final answer is: MIXTURE |
| **DECOMP** | QC: Drug effective in smoking cessation studies. 
 QS: [qa] Do the cited RCTs provide evidence of effectiveness in quitting (vs. controls) with objective verification? 
 A: Two randomized, double-blind trials with CO verification report continuous abstinence rates. 
 QS: [qa] Does the evidence establish broader effectiveness (e.g., vs. non-drug methods) or address external validity/alternatives? 
 A: It notes a lack of quantitative comparison with non-drug methods, limiting the claim's scope. 
 QS: [final_decision] What is the final label of the claim? 
 A: MIXTURE 
 QS: [EOQ] |

## APPENDIX D. FULL PROMPT TEMPLATES

Table 6: Full prompt template for Expectation–Evidence Prompting (EEP). Organized into five sections: [TD] Task Declaration, [FS] Few-shot Examples, [DI] Data Input, [SR] Structured Reasoning, and [FL] Final Label.

| Section | EEP Prompt Template |
|---|---|
| **[TD]** | You are a fact-checking agent. Use the Expectation–Evidence Prompting (EEP) strategy to analyze claims. 
 Your task is to classify each claim into one of four categories — {TRUE, FALSE, MIXTURE, UNPROVEN} — based solely on the provided evidence. |
| **[FS]** | Here are examples. 
 **Example 1** 
 Claim: Bat from Shawnee County tests positive for rabies. 
 Evidence: Topeka television station KSNT reports that the bat was found in Shawnee County. . . . survival is rare. 
 Step 1: If the claim is true, there should be evidence that a bat in Shawnee County tested positive for rabies, supported by media reports and health department advisories. 
 Step 2: If the claim is false, there should be evidence that the bat tested negative, or that the positive case involved a different animal, or that the rabid bat was not from Shawnee County. 
 Step 3: The actual evidence shows that KSNT and the county health department confirmed a rabid bat in Shawnee County, which matches Step 1 and not Step 2; therefore, the claim is true. 
 Step 4: Based on the above, the final label is: TRUE. 

 **Example 2** 
 Claim: A new Facebook feature enabling users to report problems by shaking their phones has caused . . . suspended. 
 Evidence: With little-to-no fanfare, Facebook began rolling out a new "shake-to-report" feature . . . how it works. 
 Step 1: If the claim is true, there should be evidence that the feature automatically triggers abuse reports and leads to account suspensions. 
 Step 2: If the claim is false, there should be evidence that the feature does not exist, or that it only functions as technical feedback unrelated to abuse reports or suspensions. 
 Step 3: The actual evidence shows the feature does exist but only opens a feedback form, without triggering abuse reports or suspensions, which matches Step 2 and not Step 1; therefore, the claim is false. 
 Step 4: Based on the above, the final label is: FALSE. 

 **Example 3** 
 Claim: Drug effective in smoking cessation studies. 
 Evidence: The numbers were from 2 fairly well-designed randomized, double-blind trials . . . abstinence rates. 
 Step 1: If the claim is true, there should be evidence that the drug achieved significantly higher quit rates than placebo or control groups in randomized controlled trials, and that its effectiveness was recognized. 
 Step 2: If the claim is false, there should be evidence that the drug showed no significant difference compared to placebo or control groups, that its effect was uncertain, or that it had notable adverse effects. 
 Step 3: The actual evidence shows the drug outperformed placebo in trials (matching Step 1), but lacked comparisons with non-drug methods (partly matching Step 2), so the conclusion is partial support. 
 Step 4: Based on the above, the final label is: MIXTURE. |

**Table 6 (continued)**

| Section | EEP Prompt Template |
|---|---|
| | **Example 4**
Claim: Patients should avoid taking ibuprofen to relieve pain and fever associated with COVID-19 infections.
Evidence: The French government, including Health Minister Olivier Véran, issued warnings advising ... reactions.
Step 1: If the claim is true, there should be evidence showing that ibuprofen indeed worsens COVID-19 outcomes, supported by clinical or immunological studies and endorsed in authoritative guidelines.
Step 2: If the claim is false, there should be evidence showing that ibuprofen does not worsen COVID-19, that its use makes no difference in illness duration or complication risk, and that this is confirmed by authoritative sources.
Step 3: The actual evidence consists mainly of government warnings and expert speculation, lacking conclusive research support (not meeting Step 1), and it also does not establish that ibuprofen is completely safe (not meeting Step 2); therefore, the conclusion is unproven.
Step 4: Based on the above, the final label is: UNPROVEN. |
| **[DI]** | Now consider the following case.
Claim: "{claim}"
Evidence: "{evidence}" |
| **[SR]** | Step 1: If the claim were true, what evidence should be present?
Step 2: If the claim were false, what evidence should be present?
Step 3: Compare the actual evidence to these expectations — does it fully match Step 1 (TRUE), fully match Step 2 (FALSE), contain elements matching both (MIXTURE), or match neither (UNPROVEN)? |
| **[FL]** | Step 4: Based on the above, the final label is: one of TRUE, FALSE, MIXTURE, or UNPROVEN. |

Table 7: Full prompt template for Standard (Answer-only). Organized into five sections: [TD] Task Declaration, [FS] Few-shot Examples, [DI] Data Input, [SR] Unstructured Reasoning, and [FL] Final Label.

| Section | Standard Prompt Template |
|---|---|
| **[TD]** | You are a fact-checking agent. Use the Standard Prompting (answer only) strategy to analyze claims.
Your task is to classify each claim into one of four categories — {TRUE, FALSE, MIXTURE, UNPROVEN} — based solely on the provided evidence. |
| **[FS]** | Here are examples:
**Example 1:**
Claim: Bat from Shawnee County tests positive for rabies.
Evidence: Topeka television station KSNT reports that the bat was found in Shawnee County. ... survival is rare.
Final label: TRUE
**Example 2:**
Claim: A new Facebook feature enabling users to report problems by shaking their phones has caused ... suspended.
Evidence: With little-to-no fanfare, Facebook began rolling out a new 'shake-to-report' feature ... how it works.
Final label: FALSE
**Example 3:**
Claim: Drug effective in smoking cessation studies.
Evidence: The numbers were from 2 fairly well-designed randomized, double-blind trials ... abstinence rates.
Final label: MIXTURE |

**Table 7 (continued)**

| Section | Standard Prompt Template |
|---------|--------------------------|
| | **Example 4:**
Claim: Patients should avoid taking ibuprofen to relieve pain and fever associated with COVID-19 infections.
Evidence: The French government, including Health Minister Olivier Véran, issued warnings advising ... reactions.
Final label: UNPROVEN |
| **[DI]** | Now consider the following case.
Claim: "{claim}"
Evidence: "{evidence}" |
| **[SR]** | No reasoning steps; directly output label. |
| **[FL]** | Final label: one of TRUE, FALSE, MIXTURE, or UNPROVEN. |

Table 8: Full prompt template for Chain-of-Thought (CoT). Organized into five sections: [TD] Task Declaration, [FS] Few-shot Examples, [DI] Data Input, [SR] Unstructured Reasoning, and [FL] Final Label.

| Section | CoT Prompt Template |
|---------|---------------------|
| **[TD]** | You are a fact-checking agent. Use the Chain-of-Thought (CoT) strategy to analyze claims.
Your task is to classify each claim into one of four categories — {TRUE, FALSE, MIXTURE, UNPROVEN} — based solely on the provided evidence. |
| **[FS]** | Here are examples:
**Example 1:**
Claim: Bat from Shawnee County tests positive for rabies.
Evidence: Topeka television station KSNT reports that the bat was found in Shawnee County. ... survival is rare.
Reasoning: The evidence shows a bat in Shawnee County linked to rabies warnings, which supports the claim without contradiction.
Final label: TRUE
**Example 2:**
Claim: A new Facebook feature enabling users to report problems by shaking their phones has caused ... suspended.
Evidence: With little-to-no fanfare, Facebook began rolling out a new 'shake-to-report' feature ... how it works.
Reasoning: The evidence confirms the feature but directly contradicts the claim about accidental abuse reports and suspensions.
Final label: FALSE
**Example 3:**
Claim: Drug effective in smoking cessation studies.
Evidence: The numbers were from 2 fairly well-designed randomized, double-blind trials ... abstinence rates.
Reasoning: The evidence supports drug effectiveness relative to other drugs and placebo but lacks comparison to non-drug methods, so support is partial.
Final label: MIXTURE
**Example 4:**
Claim: Patients should avoid taking ibuprofen to relieve pain and fever associated with COVID-19 infections.
Evidence: The French government, including Health Minister Olivier Véran, issued warnings advising ... reactions.
Reasoning: The warnings exist, but the evidence is not conclusive and lacks scientific consensus.
Final label: UNPROVEN |

**Table 8 (continued)**

| Section | CoT Prompt Template |
|---------|---------------------|
| **[DI]** | Now consider the following case.
Claim: "{claim}"
Evidence: "{evidence}" |
| **[SR]** | Reasoning: |
| **[FL]** | Final label: one of TRUE, FALSE, MIXTURE, or UNPROVEN. |

Table 9: Full prompt template for Self-Ask. Organized into five sections: [TD] Task Declaration, [FS] Few-shot Examples, [DI] Data Input, [SR] Unstructured Reasoning, and [FL] Final Label.

| Section | Self-Ask Prompt Template |
|---------|--------------------------|
| **[TD]** | You are a fact-checking agent. Use the Self-Ask prompting strategy to analyze claims. Your task is to classify each claim into one of four categories — {TRUE, FALSE, MIXTURE, UNPROVEN} — based solely on the provided evidence. |
| **[FS]** | Here are examples:
**Example 1:**
Claim: Bat from Shawnee County tests positive for rabies.
Evidence: Topeka television station KSNT reports that the bat was found in Shawnee County. ... survival is rare.
Are follow up questions needed here: Yes.
Follow up: Does the evidence describe whether the bat found in Shawnee County was confirmed rabid?
Intermediate answer: The evidence reports that the bat found in Shawnee County tested positive for rabies.
So the final answer is: TRUE
**Example 2:**
Claim: A new Facebook feature enabling users to report problems by shaking their phones has caused ... suspended.
Evidence: With little-to-no fanfare, Facebook began rolling out a new 'shake-to-report' feature ... how it works.
Are follow up questions needed here: Yes.
Follow up: How does the evidence describe the existence of this feature?
Intermediate answer: The evidence says the "shake-to-report" feature is real and is automatically enabled in the app.
Follow up: How does the evidence describe its relation to abuse reports or account suspensions?
Intermediate answer: The evidence explains it only opens a feedback form for technical problems and cannot trigger abuse reports or suspensions.
So the final answer is: FALSE
**Example 3:**
Claim: Drug effective in smoking cessation studies.
Evidence: The numbers were from 2 fairly well-designed randomized, double-blind trials ... abstinence rates.
Are follow up questions needed here: Yes.
Follow up: How does the evidence indicate the drug's effectiveness?
Intermediate answer: The evidence shows higher continuous abstinence rates in the drug groups compared with placebo, indicating effectiveness.
Follow up: Does the evidence also provide comparisons with non-drug interventions?
Intermediate answer: The evidence states that no quantitative comparison with non-drug methods was included.
So the final answer is: MIXTURE |

**Table 9 (continued)**

| Section | Self-Ask Prompt Template |
|---|---|
| | **Example 4:**
Claim: Patients should avoid taking ibuprofen to relieve pain and fever associated with COVID-19 infections.
Evidence: The French government, including Health Minister Olivier Véran, issued warnings advising ... reactions.
Are follow up questions needed here: Yes.
Follow up: What positions do the government and some doctors take in the evidence?
Intermediate answer: The French government and some doctors advised against ibuprofen, expressing concern it could worsen illness.
Follow up: What does the evidence say about the scientific consensus?
Intermediate answer: The evidence shows the medical community was divided, with some experts saying there was no solid evidence to support the warning.
So the final answer is: UNPROVEN |
| **[DI]** | Now consider the following case.
Question: "{claim}"
Evidence: "{evidence}" |
| **[SR]** | Are follow up questions needed here: Yes/No
[If YES] Follow up:
Intermediate answer: [Answer based on the evidence]
[Optional if still undecidable] Follow up:
Intermediate answer: [Answer based on the evidence]
[Optional if still undecidable] Follow up:
Intermediate answer: [Answer based on the evidence]
... |
| **[FL]** | So the final answer is: one of TRUE, FALSE, MIXTURE, or UNPROVEN. |

Table 10: Full prompt template for Decompose-Then-Reason (DE-COMP). Organized into five sections: [TD] Task Declaration, [FS] Few-shot Examples, [DI] Data Input, [SR] Structured Reasoning, and [FL] Final Label.

| Section | DECOMP Prompt Template |
|---|---|
| **[TD]** | You are a fact-checking agent. Use the Decompose-Then-Reason (DECOMP) strategy to analyze claims.
Your task is to classify each claim into one of four categories — {TRUE, FALSE, MIXTURE, UNPROVEN} — based solely on the provided evidence. |
| **[FS]** | Here are examples:
**Example 1:**
Claim: Bat from Shawnee County tests positive for rabies.
Evidence: Topeka television station KSNT reports that the bat was found in Shawnee County. ... survival is rare.
QC: Bat from Shawnee County tests positive for rabies.
QS: [qa] Does the evidence link the rabies case to a bat from Shawnee County? A: It cites a KSNT report about a bat found in Shawnee County and a county health advisory consistent with a confirmed rabid bat.
QS: [qa] Does the evidence indicate a positive rabies test/result for that bat? A: The context (news report + public health warning about a rabies case) indicates the bat tested positive.
QS: [final_decision] What is the final label of the claim? A: TRUE
QS: [EOQ] |

Continued on next page

**Table 10 (continued)**

| Section | DECOMP Prompt Template |
|---|---|
| | **Example 2:**
Claim: A new Facebook feature enabling users to report problems by shaking their phones has caused ... suspended.
Evidence: With little-to-no fanfare, Facebook began rolling out a new 'shake-to-report' feature ... how it works.
QC: A new Facebook feature enabling users to report problems by shaking their phones has caused ... suspended.
QS: [qa] Does the evidence say shake-to-report files abuse reports automatically? A: It opens a form and requires user input; accidental reports are not sent.
QS: [qa] Does the evidence say the feature can suspend accounts or trigger abuse reports inadvertently? A: It explicitly says that's not how it works and it's for "something isn't working," not abuse.
QS: [final_decision] What is the final label of the claim? A: FALSE
QS: [EOQ] |
| | **Example 3:**
Claim: Drug effective in smoking cessation studies.
Evidence: The numbers were from 2 fairly well-designed randomized, double-blind trials ... abstinence rates.
QC: Drug effective in smoking cessation studies.
QS: [qa] Do the cited RCTs provide evidence of effectiveness in quitting (vs. controls) with objective verification? A: Two randomized, double-blind trials with CO verification report continuous abstinence rates.
QS: [qa] Does the evidence establish broader effectiveness (e.g., vs. non-drug methods) or address external validity/alternatives? A: It notes a lack of quantitative comparison with non-drug methods, limiting the claim's scope.
QS: [final_decision] What is the final label of the claim? A: MIXTURE
QS: [EOQ] |
| | **Example 4:**
Claim: Patients should avoid taking ibuprofen to relieve pain and fever associated with COVID-19 infections.
Evidence: The French government, including Health Minister Olivier Véran, issued warnings advising ... reactions.
QC: Patients should avoid taking ibuprofen to relieve pain and fever associated with COVID-19 infections.
QS: [qa] Does the evidence show authoritative warnings advising against ibuprofen/NSAIDs for COVID-19? A: French officials issued such warnings.
QS: [qa] Does the evidence establish strong scientific proof of harm or a confirmed causal risk? A: It explicitly notes mixed reactions and that some say scientific evidence was lacking.
QS: [final_decision] What is the final label of the claim? A: UNPROVEN
QS: [EOQ] |
| **[DI]** | Now consider the following case.
Claim: "{claim}"
Evidence: "{evidence}" |
| **[SR]** | QC: {claim}
QS: [qa] Question 1 (derived from QC, only if needed)
A: Answer 1 (extracted from Evidence)
QS: [qa] Question 2 (derived from QC, only if needed)
A: Answer 2 (extracted from Evidence)
... |
| **[FL]** | QS: [final_decision] Based on all the above, what is the final label of the claim?
A: one of TRUE, FALSE, UNPROVEN, or MIXTURE.
QS: [EOQ] |

# APPENDIX E. FULL CLASSIFICATION REPORTS

Table 11: Classification report (EEP) on the evaluation set (N=1998). Macro average is the unweighted mean across classes. Weighted average is the mean weighted by class support.

| Class | Precision | Recall | F1-score | Support |
|---|---|---|---|---|
| NOT ENOUGH INFO | 0.794 | 0.995 | 0.883 | 666 |
| SUPPORTS | 0.890 | 0.914 | 0.902 | 666 |
| REFUTES | 0.960 | 0.691 | 0.803 | 666 |
| **Accuracy** | | **0.867** | | **1998** |
| **Macro avg** | 0.882 | 0.867 | 0.863 | 1998 |
| **Weighted avg** | 0.882 | 0.867 | 0.863 | 1998 |

Table 12: Classification report (Chain-of-Thought) on the evaluation set (N=1998). Macro average is the unweighted mean across classes. Weighted average is the mean weighted by class support.

| Class | Precision | Recall | F1-score | Support |
|---|---|---|---|---|
| NOT ENOUGH INFO | 0.814 | 0.991 | 0.894 | 666 |
| REFUTES | 0.953 | 0.572 | 0.715 | 666 |
| SUPPORTS | 0.804 | 0.950 | 0.871 | 666 |
| **Accuracy** | | **0.838** | | **1998** |
| **Macro avg** | 0.857 | 0.838 | 0.827 | 1998 |
| **Weighted avg** | 0.857 | 0.838 | 0.827 | 1998 |

Table 13: Classification report (Self-Ask) on the evaluation set (N=1998). Macro average is the unweighted mean across classes. Weighted average is the mean weighted by class support.

| Class | Precision | Recall | F1-score | Support |
|---|---|---|---|---|
| NOT ENOUGH INFO | 0.721 | 0.916 | 0.807 | 666 |
| SUPPORTS | 0.849 | 0.859 | 0.854 | 666 |
| REFUTES | 0.879 | 0.631 | 0.734 | 666 |
| **Accuracy** | | **0.802** | | **1998** |
| **Macro avg** | 0.816 | 0.802 | 0.798 | 1998 |
| **Weighted avg** | 0.816 | 0.802 | 0.798 | 1998 |

Table 14: Classification report (Decompose-Then-Reason) on the evaluation set (N=1998). Macro average is the unweighted mean across classes. Weighted average is the mean weighted by class support.

| Class | Precision | Recall | F1-score | Support |
|---|---|---|---|---|
| NOT ENOUGH INFO | 0.688 | 1.000 | 0.815 | 666 |
| REFUTES | 0.972 | 0.530 | 0.686 | 666 |
| SUPPORTS | 0.903 | 0.904 | 0.903 | 666 |
| **Accuracy** | | **0.811** | | **1998** |
| **Macro avg** | 0.854 | 0.811 | 0.802 | 1998 |
| **Weighted avg** | 0.854 | 0.811 | 0.802 | 1998 |

Table 15: Classification report (EEP) on the full FEVER dev set (N=19998). Macro average is the unweighted mean across classes. Weighted average is the mean weighted by class support.

| Class | Precision | Recall | F1-score | Support |
|---|---|---|---|---|
| SUPPORTS | 0.888 | 0.924 | 0.906 | 6666 |
| REFUTES | 0.969 | 0.701 | 0.813 | 6666 |
| NOT ENOUGH INFO | 0.804 | 0.994 | 0.889 | 6666 |
| **Accuracy** | | **0.873** | | **19998** |
| **Macro avg** | 0.887 | 0.873 | 0.869 | 19998 |
| **Weighted avg** | 0.887 | 0.873 | 0.869 | 19998 |

Table 16: Classification report (Expectation–Evidence Prompting) on the PubHealth evaluation set (N=1214). Macro average is the unweighted mean across classes. Weighted average is the mean weighted by class support.

| Class | Precision | Recall | F1-score | Support |
|---|---|---|---|---|
| FALSE | 0.871 | 0.532 | 0.660 | 380 |
| MIXTURE | 0.306 | 0.543 | 0.391 | 164 |
| TRUE | 0.971 | 0.749 | 0.846 | 629 |
| UNPROVEN | 0.117 | 0.585 | 0.194 | 41 |
| **Accuracy** | | **0.647** | | **1214** |
| **Macro avg** | 0.566 | 0.602 | 0.523 | 1214 |
| **Weighted avg** | 0.821 | 0.647 | 0.704 | 1214 |

Table 17: Classification report (Standard) on the PubHealth evaluation set (N=1214). Macro average is the unweighted mean across classes. Weighted average is the mean weighted by class support.

| Class | Precision | Recall | F1-score | Support |
|---|---|---|---|---|
| FALSE | 0.816 | 0.629 | 0.710 | 380 |
| MIXTURE | 0.295 | 0.744 | 0.423 | 164 |
| TRUE | 0.967 | 0.647 | 0.775 | 629 |
| UNPROVEN | 0.138 | 0.293 | 0.187 | 41 |
| **Accuracy** | | **0.643** | | **1214** |
| **Macro avg** | 0.554 | 0.578 | 0.524 | 1214 |
| **Weighted avg** | 0.801 | 0.643 | 0.687 | 1214 |

Table 18: Classification report (Chain-of-Thought) on the PubHealth evaluation set (N=1214). Macro average is the unweighted mean across classes. Weighted average is the mean weighted by class support.

| Class | Precision | Recall | F1-score | Support |
|---|---|---|---|---|
| FALSE | 0.793 | 0.634 | 0.705 | 380 |
| MIXTURE | 0.298 | 0.628 | 0.404 | 164 |
| TRUE | 0.951 | 0.652 | 0.774 | 629 |
| UNPROVEN | 0.083 | 0.268 | 0.126 | 41 |
| **Accuracy** | | **0.630** | | **1214** |
| **Macro avg** | 0.531 | 0.546 | 0.502 | 1214 |
| **Weighted avg** | 0.784 | 0.630 | 0.680 | 1214 |

Table 19: Classification report (Self-Ask) on the PubHealth evaluation set (N=1214). Macro average is the unweighted mean across classes. Weighted average is the mean weighted by class support.

| Class | Precision | Recall | F1-score | Support |
|---|---|---|---|---|
| FALSE | 0.758 | 0.650 | 0.700 | 380 |
| MIXTURE | 0.232 | 0.439 | 0.303 | 164 |
| TRUE | 0.946 | 0.499 | 0.653 | 629 |
| UNPROVEN | 0.065 | 0.390 | 0.112 | 41 |
| **Accuracy** | | **0.535** | | **1214** |
| **Macro avg** | 0.500 | 0.495 | 0.442 | 1214 |
| **Weighted avg** | 0.761 | 0.535 | 0.602 | 1214 |

Table 20: Classification report (Decompose-Then-Reason) on the PubHealth evaluation set (N=1214). Macro average is the unweighted mean across classes. Weighted average is the mean weighted by class support.

| Class | Precision | Recall | F1-score | Support |
|---|---|---|---|---|
| FALSE | 0.793 | 0.684 | 0.734 | 380 |
| MIXTURE | 0.345 | 0.463 | 0.396 | 164 |
| TRUE | 0.975 | 0.693 | 0.810 | 629 |
| UNPROVEN | 0.100 | 0.537 | 0.169 | 41 |
| **Accuracy** | | **0.654** | | **1214** |
| **Macro avg** | 0.553 | 0.594 | 0.527 | 1214 |
| **Weighted avg** | 0.804 | 0.654 | 0.709 | 1214 |

Table 21: Classification report (Expectation–Evidence Prompting) on the SciFact evaluation set (N=452). Macro average is the unweighted mean across classes. Weighted average is the mean weighted by class support.

| Class | Precision | Recall | F1-score | Support |
|---|---|---|---|---|
| NOINFO | 0.576 | 0.728 | 0.643 | 114 |
| REFUTES | 0.753 | 0.500 | 0.601 | 122 |
| SUPPORTS | 0.736 | 0.773 | 0.754 | 216 |
| **Accuracy** | | **0.688** | | **452** |
| **Macro avg** | 0.688 | 0.667 | 0.666 | 452 |
| **Weighted avg** | 0.700 | 0.688 | 0.685 | 452 |

Table 22: Classification report (Standard) on the SciFact evaluation set (N=452). Macro average is the unweighted mean across classes. Weighted average is the mean weighted by class support.

| Class | Precision | Recall | F1-score | Support |
|---|---|---|---|---|
| NOINFO | 0.607 | 0.149 | 0.239 | 114 |
| REFUTES | 0.641 | 0.336 | 0.441 | 122 |
| SUPPORTS | 0.553 | 0.921 | 0.691 | 216 |
| **Accuracy** | | **0.569** | | **452** |
| **Macro avg** | 0.600 | 0.469 | 0.457 | 452 |
| **Weighted avg** | 0.590 | 0.569 | 0.510 | 452 |

Table 23: Classification report (Chain-of-Thought) on the SciFact evaluation set (N=452). Macro average is the unweighted mean across classes. Weighted average is the mean weighted by class support.

| Class | Precision | Recall | F1-score | Support |
|---|---|---|---|---|
| NOINFO | 0.567 | 0.816 | 0.669 | 114 |
| REFUTES | 0.760 | 0.598 | 0.670 | 122 |
| SUPPORTS | 0.807 | 0.718 | 0.760 | 216 |
| **Accuracy** | | **0.710** | | **452** |
| **Macro avg** | 0.712 | 0.711 | 0.700 | 452 |
| **Weighted avg** | 0.734 | 0.710 | 0.713 | 452 |

Table 24: Classification report (Self-Ask) on the SciFact evaluation set (N=452). Macro average is the unweighted mean across classes. Weighted average is the mean weighted by class support.

| Class | Precision | Recall | F1-score | Support |
|---|---|---|---|---|
| NOINFO | 0.431 | 0.789 | 0.557 | 114 |
| REFUTES | 0.784 | 0.475 | 0.592 | 122 |
| SUPPORTS | 0.757 | 0.593 | 0.665 | 216 |
| **Accuracy** | | **0.611** | | **452** |
| **Macro avg** | 0.657 | 0.619 | 0.605 | 452 |
| **Weighted avg** | 0.682 | 0.611 | 0.618 | 452 |

Table 25: Classification report (Decompose-Then-Reason) on the SciFact evaluation set (N=452). Macro average is the unweighted mean across classes. Weighted average is the mean weighted by class support.

| Class | Precision | Recall | F1-score | Support |
|---|---|---|---|---|
| NOINFO | 0.753 | 0.509 | 0.607 | 114 |
| REFUTES | 0.800 | 0.328 | 0.465 | 122 |
| SUPPORTS | 0.609 | 0.917 | 0.732 | 216 |
| **Accuracy** | | **0.655** | | **452** |
| **Macro avg** | 0.721 | 0.584 | 0.601 | 452 |
| **Weighted avg** | 0.697 | 0.655 | 0.629 | 452 |