# OpenReview forum: "Expectation–Evidence Prompting: Structuring Verification by Comparing Expected and Observed Evidence"
_ICLR.cc/2026/Conference — ICLR 2026 Conference Withdrawn Submission_

### Official Review · Reviewer_KAAi · 2025-10-23

**Soundness:** 2
**Presentation:** 3
**Contribution:** 2
**Rating:** 4
**Confidence:** 4

**Summary:**

This paper introduces a prompting framework, named Expectation–Evidence Prompting (EEP), for large language models to enhance factual verification. Drawing from the Strategic Use of Evidence technique in cognitive psychology, EEP involves generating two sets of expectations, supportive and refutational, and comparing them to observed evidence using a semantic consistency function. The framework is also extended to a supervised learning setup with cross-entropy loss and regularization. Evaluated on three benchmarks using GPT-3.5-turbo, EEP outperforms baselines like Chain-of-Thought, Self-Ask, and Decompose.

**Strengths:**

1. Exploring reasoning and verification strategies from a cognitive perspective is promising, and the proposed EEP framework is reasonable.

2. The proposed method is easy to follow, and the prompt templates used in this paper are very clear.

**Weaknesses:**

1. I personally think the innovation is limited, as the idea of combining supportive and refutational evidence has already appeared in ArgLLM[1].

2. The generalization ability of the proposed method requires further verification. It only uses GPT-3.5 as the base model and lacks experiments on other closed-source and open-source models. For complex claims such as those in WiCE[2] or HoVer[3], is the proposed method still effective?

3. The paper lacks implementation details of the evidence retrieval component. In Figure 3, are the pieces of evidence used for supportive and refutational queries the same?

4. The baselines compared in the paper are too weak. It lacks comparisons with recent works [4, 5], making the results unconvincing.

5. The proposed supervised learning method does not seem very effective, and comparing it with unsupervised baselines is unfair.

[1] Freedman G, Dejl A, Gorur D, et al. Argumentative Large Language Models for Explainable and Contestable Claim Verification[C]//Proceedings of the AAAI Conference on Artificial Intelligence. 2025, 39(14): 14930-14939.

[2] Kamoi R, Goyal T, Rodriguez J D, et al. Wice: Real-world entailment for claims in wikipedia[J]. arXiv preprint arXiv:2303.01432, 2023.

[3] Jiang Y, Bordia S, Zhong Z, et al. HoVer: A dataset for many-hop fact extraction and claim verification[J]. arXiv preprint arXiv:2011.03088, 2020.

[4] Zhao R, Flanigan J. SYNTHVERIFY: Enhancing Zero-Shot Claim Verification through Step-by-Step Synthetic Data Generation[C]//Findings of the Association for Computational Linguistics: ACL 2025. 2025: 3257-3274.

[5] Lu Y, Ziems N, Dang H, et al. Optimizing decomposition for optimal claim verification[J]. arXiv preprint arXiv:2503.15354, 2025.

**Questions:**

please address weaknesses

---

### Official Review · Reviewer_6awL · 2025-10-30

**Soundness:** 3
**Presentation:** 3
**Contribution:** 2
**Rating:** 4
**Confidence:** 3

**Summary:**

This paper introduces Expectation–Evidence Prompting (EEP), a cognitive science inspired framework for factual verification in large language models (LLMs). Instead of directly mapping claims to truth labels, EEP guides the model to generate supportive and refutational expectations about what evidence should exist if a claim were true or false. These expectations are then compared to observed evidence using a semantic consistency function, producing support and refutation scores. This is evaluated with a variety of methods including Implicit LLM reasoning, embedding similarity and Natural Language Inference. A claim is accepted, rejected, or abstained from based on thresholded scores. The authors motivate EEP with parallels to the Strategic Use of Evidence (SUE) technique in investigative psychology and evaluate it on FEVER, PubHealth, and SciFact. EEP achieves competitive results, notably 86.3 macro-F1 on FEVER (+3.6 over CoT), 82.1 precision on PubHealth, and 76.1 F1 on the SUPPORTS class in SciFact. EEP thus formalizes a bidirectional reasoning mechanism that improves interpretability and robustness compared to Chain-of-Thought (CoT), Self-Ask, and DECOMP prompting.

**Strengths:**

Introduces a structured, cognitively grounded prompting framework that imposes logical symmetry and transparency. Some novelty in bridging psychological theory and LLM prompting.

Flexible implementation: can use LLM-based entailment, embedding similarity, or NLI-based semantic scoring.

Strong empirical results, especially in precision and interpretability, with clear ablations and reproducible templates.

Provides a principled abstention mechanism instead of ad hoc uncertainty thresholds.

**Weaknesses:**

The psychological analogy (SUE) is conceptually interesting but only loosely validated in the LLM context; its cognitive plausibility may not imply computational efficacy.

Performance gains are modest, raising questions about the significance of improvements relative to model variance.

Lacks direct comparison to structured reasoning frameworks (e.g., Tree-of-Thought or Graph-of-Thought) that also model entailment and evidence relationships.

Thresholding mechanisms are not fully or apparently justified, it is unclear whether they were tuned on validation or derived heuristically.

Limited exploration of cross-domain generalization or impact of expectation quality on downstream reasoning robustness.

**Questions:**

Why were maximum support/refutation scores used instead of averages or other aggregations? Would this reduce robustness to noisy evidence?

Are the reported F1 improvements statistically significant given the small magnitude?

How were thresholds determined for non-trainable variants?

Which semantic consistency variant (LLM judgment, embedding, or NLI) was used in the main experiments? Were cross-method results compared?

How does EEP compare against structured reasoning frameworks like Tree-of-Thought or Graph-of-Thought that also model entailment relationships?

---

### Official Review · Reviewer_VWVY · 2025-10-31

**Soundness:** 2
**Presentation:** 3
**Contribution:** 2
**Rating:** 4
**Confidence:** 3

**Summary:**

This paper introduces Expectation–Evidence Prompting (EEP), a cognitively inspired prompting framework for factual verification with large language models (LLMs). Drawing on the Strategic Use of Evidence (SUE) technique from cognitive psychology, EEP prompts the LLM to generate both supportive and refutational expectations for a claim, then explicitly compares these with observed evidence to make a structured three-way decision: support, refute, or abstain. The method is evaluated on three standard fact-checking benchmarks (FEVER, PubHealth, SciFact) and compared to strong prompting baselines (Standard, Chain-of-Thought, Self-Ask, DECOMP). EEP achieves state-of-the-art macro-F1 on FEVER and strong precision on PubHealth, with consistent gains in main metrics.

**Strengths:**

- The authors propose EEP, which is conceptually interesting and well-motivated, bridging cognitive psychology (SUE) and structured LLM prompting for factual verification.
- The proposed method is clear, modular, and interpretable, which enforces bidirectional reasoning and provides a principled abstention mechanism.
- Experiment results show the effectiveness of the proposed method, with evaluation on three representative benchmarks, with strong baselines and detailed reporting.

**Weaknesses:**

- Technical novelty is primarily in prompt engineering.
- Evaluation is limited to GPT-3.5 Turbo; generalization to other LLMs like open-source LLMs is untested.
- No ablation on key design choices, such as semantic consistency function, thresholding, and expectation types.

**Questions:**

- How sensitive is EEP to the choice of thresholds for support/refute/abstain, and is there a principled way to set them?
- Could EEP be hybridized with decomposition-based approaches (e.g., DECOMP), and what would be the challenges or benefits of such a combination?
- How robust is EEP to poor or incorrectly generated expectations, especially for challenging or ambiguous claims? Can the authors provide a breakdown of errors due to expectation generation versus evidence alignment?

---

### Note · Authors · 2026-01-06

I have read and agree with the venue's withdrawal policy on behalf of myself and my co-authors.